# Seed dressing with mefenpyr-diethyl as a safener for mesosulfuron-methyl application in wheat: The evaluation and mechanisms

**Libing Yuan**[1,2]*, **Guangyuan Ma**[2], **Yaling Geng**[2], **Xiaomin Liu**[3], **Hua Wang**[2], **Jian Li**[2], **Shanshan Song**[2], **Wenliang Pan**[2], **Zhiying Hun**[2]

**1** College of Plant Protection, Hebei Agricultural University, Baoding, Hebei, China, **2** Hebei Academy of Agricultural and Forestry Sciences/IPM Centre of Hebei Province/Key Laboratory of Integrated Pest Management on Crops in Northern Region of North China, Plant Protection Institute, Ministry of Agriculture and Rural Affairs, Baoding, Hebei, China, **3** Cereal and Oil Crops Institute, Hebei Academy of Agriculture and Forestry Sciences, Shijiazhuang, Hebei, China

* yuanlibing83@163.com

**Data Availability Statement:** All the raw Illumina sequence reads are available from the NCBI Sequence Read Archive (SRA) database with accession number SRP263381

## Abstract

Mesosulfuron-methyl is always applied by foliar spraying in combination with the safener mefenpyr-diethyl to avoid phytotoxicity on wheat (*Triticum aestivum* L.) cultivars. However, it was observed that the tolerance of Tausch's goatgrass (*Aegilops tauschii* Coss.) to mesosulfuron-methyl significantly increased in the presence of mefenpyr-diethyl by performing bioassay. This confirmed phenomenon may lead to overuse of mesosulfuron-methyl and weed resistance evolution in field conditions. Therefore, we tested the effect of wheat seed dressing with mefenpyr-diethyl as a possible alternative and disclosed the underlying mechanisms by herbicide dissipation study, enzymatic analysis and transcriptome profiling. The results suggest that increase of ALS activity, enhancement of metabolic processes, and other stress responses are crucial for the regulation of herbicide detoxification induced by mefenpyr-diethyl. Additionally, transcription factors such as *AP2/ERF-ERF*, *bHLH*, *NAC*, and *MYB*, and protein kinase such as RLK-Pelle_DLSV might play vital regulatory roles. The current study has important implications for mesosulfuron-methyl application in wheat field to control Tausch's goatgrass and provides a comprehensive understanding of the protective effect of mefenpyr-diethyl.

## Introduction

Tausch's goatgrass (*Aegilops tauschii* Coss.) is one of the most troublesome weeds in winter wheat (*Triticum aestivum* L.) fields in China [1]. It could cause 50–80% yield loss in wheat producing regions [2,3]. Most herbicides are poor in selectivity due to the similarity of Tausch's goatgrass and wheat—Tausch's goatgrass is the D-genome progenitor of hexaploid wheat and has parallel growth habits with wheat [4]. So far, the sulfonylurea herbicide mesosulfuron-methyl is the most acceptable herbicide for controlling Tausch's goatgrass in wheat field.

Mesosulfuron-methyl interferes with the biosynthetic pathway of branched-chain amino acids by inhibiting the activity of acetolactate synthase (ALS), which is also known as

**Funding:** This study was supported by Key R&D Program of Hebei Province (18226511), National Key R&D Program of China (2016YFD0300705) and HAAFS Agriculture Science and Technology Innovation Project (2019-1-1-1). The funders had no role in study design, data collection and analysis, decision to publish, or preparation of the manuscript.

**Competing interests:** The authors have declared that no competing interests exist.

acetohydroxy acid synthase (AHAS) [5,6]. It is a sulfonylurea herbicide developed for post-emergence control of a wide spectrum of grasses and some broad-leaved weeds in wheat field [7,8]. It is always applied by foliar spraying in combination with the safener mefenpyr-diethyl, which protects cereal crops from adverse effect [9]. There are some studies showing that some safeners including mefenpyr-diethyl could increase the tolerance of weeds to herbicides [10–12]. As yet, however, there has been no report suggesting that mefenpyr-diethyl could increase the tolerance of Tausch's goatgrass to mesosulfuron-methyl while protecting wheat from damages. If such is the case, then spraying of mefenpyr-diethyl may result in mesosulfuron-methyl overdose and weed resistance evolution.

Enhancement of herbicide metabolism is the main detoxification mechanisms for safeners to protect plants [13]. This process generally involves Cytochrome P450 monooxygenase (CYP450), glutathione S-transferase (GST), glucosyltransferases, and ATP-binding cassette transporters (ABCs) [14], etc. However, activity differences among these enzymes are not sufficient to explain the different responses among different plant species to herbicides and safeners. The investigation into the genes endowing plants with these traits is still in progress [15]. The development of transcriptome techniques in recent years facilitates us to identify more elements participating in the response to herbicide and safeners. Nevertheless, Das *et al.* pointed out that transcriptional signatures were quite different across species and even herbicides having the same target enzyme or similar chemical structures caused differentiate alterations in gene sets [16]. Despite that the action mode of mesosulfuron-methyl and selectivity of mefenpyr-diethyl have been clearly elucidated [17], there is no information on the precise molecular mechanisms for their phytotoxic and detox effect.

The present study seeks to examine whether mefenpyr-diethyl spraying would increase the tolerance of Tausch's goatgrass to mesosulfuron-methyl and test the seed dressing method as an alternative. The dissipation rate of mesosulfuron-methyl in wheat and Tausch's goatgrass and dose responses to mesosulfuron-methyl were compared. By detecting ALS activity, CYP450 and GST content, and employing transcriptome profiling, we attempt to illuminate the response of wheat to these chemicals and the underlying mechanisms. This work will contribute to the control of Tausch's goatgrass and facilitate our understanding of interrelationship between wheat and herbicides/safeners.

## Material and methods

### Plant materials and chemicals

Seeds of Tausch's goatgrass and wheat were randomly collected from at least 200 individual plants distributed in the winter wheat field in Nanhe County, Hebei province, China (37˚ 01'45.6" N; 114˚41'34.1" E). The wheat variety was Jimai22. The sampled field was under a repeated wheat-corn rotation for several decades. Mesosulfuron-methyl was applied only in the former year. No suspected resistance to mesosulfuron-methyl of Tausch's goatgrass was found according to the land owner. The seeds were planted in potting mix comprising 1:1 (v/v) peat and sand in 7 cm radius pots (20 seeds per pot). These pots were kept at 20˚C in a 12:12 h light/dark cycle and watered as required. Mesosulfuron-methyl and mefenpyr-diethyl were purchased from Shandong Binnong Technology Co., Ltd. and Jiangsu Tianrong Group Co., Ltd. respectively.

### Whole-plant bioassay

Seedlings of Tausch's goatgrass and wheat at 1-leaf stage were selected for whole-plant bioassay. The samples were divided into three groups: plants treated with mesosulfuron-methyl by spraying (Mmsp), plants treated with mesosulfuron-methyl and mefenpyr-diethyl by spraying

**Table 1. Doses of mesosulfuron-methyl applied in each group*.**

|  | Tausch's goatgrass | wheat |
|---|---|---|
| **Mmsp** | 0, 1/1280, 1/128, 1/64, 1/32, 1/16, 1/8, 1.25 | 0, 1/80, 1/8, 1/4, 1/2, 1, 2, 20 |
| **MmMdsp** | 0, 1/160, 1/16, 1/8, 1/4, 1/2, 1, 10 | 0, 1/20, 1/2, 1, 2, 4, 8, 80 |
| **MmMdsd** | - | 0, 1/5, 2, 4, 8, 16, 32, 160 |

* Doses of mesosulfuron-methyl in the table were indicated as the fold of the field recommended dose 13.5 g ai ha$^{-1}$.

(MmMdsp), plants treated with mesosulfuron-methyl by spaying and pretreated with mefenpyr-diethyl by seed dressing (MmMdsd). The doses of mesosulfuron-methyl applied in each group were listed in Table 1. Mefenpyr-diethyl was applied at 27 g ai ha$^{-1}$ in MmMdsp group. For MmMdsd, based on preliminary experiments, 2 g per kg seed was chosen for mefenpyr-diethyl seed dressing (S1 Table). Solutions were sprayed using a research track sprayer (3WP-2000) which delivered 450 L ha$^{-1}$ spray solution at 0.3 MPa. Herbicide treatments were arranged in a completely randomized design with three replications, and the experiment was conducted twice over time with identical experimental procedures. Seven days after treatment, seedlings were harvested, and the plant height was measured [18–22].

## Dissipation of mesosulfuron-methyl

Three replicate samples of 30 plants (except roots) were harvested at 0, 1/8, 1/4, 1/2, 1, 2, 3, 5, 7, 10, 15, and 20 days after treatment (DAT) and stored at −20˚C for further studies. Mesosulfuron-methyl extraction, residue analysis, and LC-MS/MS method validation were carried out according to Zhao *et al*. with some modifications [23]. Briefly, plant tissue was grounded into powder in liquid nitrogen and suspended in methyl cyanides (MeCN) containing 1% (v/v) formic acid. After adding 2 g of NaCl, the mixture was shaken for 2 min before centrifugation at 5000 rpm for 5 min, after which 50 mg of primary secondary amine (PSA), 150 mg of anhydrous magnesium sulphate ($MgSO_4$), and 2 mg of graphitized carbon black (GCB) were added to the supernatant. The tube was then shaken for another 1 min and centrifuged at 12,000 rpm for 5 min. The final supernatant was filtered through a 0.22 μm membrane and used for LC-MS/MS analysis.

LC was performed using a Dionex Ultimate 3000 LC system (Thermo fishier, San Jose, USA). The MS/MS analysis was performed using a Thermo-Finnigan (TSQ Quantum Ultra, San Jose, CA, USA) triple-quadrupole mass spectrometer equipped with a Z-Spray™ electrospray ionization (ESI) source (Waters) which was coupled online to the UPLC system and operated in the multiple reactions monitoring (MRM) mode. Mesosulfuron-methyl levels in each sample were quantified using a standard curve. MassLynx 4.1 software was used to collect and analyze the obtained data.

## ALS activity, CYP450 content, and GST content assay

Wheat was treated with 1.69 g ai ha$^{-1}$ mesosulfuron-methyl at 1-leaf stage (Mm), seed-dressed with 2 g mefenpyr-diethyl per kg seed (Md), or seed-dressed with 2 g mefenpyr-diethyl per kg seed and then treated with 1.69 g ai ha$^{-1}$ mesosulfuron-methyl at 1-leaf stage (MmMd). Wheat seedlings without any treatment were set as control (CK). ALS activity, CYP450 content and GST content assay were conducted 1, 3, 5, 7 days after mesosulfuron-methyl application. ALS activity was assayed according to Simpson *et al*. [24]. The experiment was conducted twice and all treatments were replicated three times.

The content of CYP450 and GST was determined using the enzyme-linked immunosorbent assay (ELISA) kits purchased from Beijing lvyuandade biotechnology Co., Ltd. following the instructions.

## Transcriptome analysis

Wheat seedlings of 5 days after mesosulfuron-methyl spraying were randomly selected for transcriptome analysis. The samples were frozen in liquid nitrogen and stored at -80˚C until analysis. Total RNA was isolated using Trizol reagent (TaKaRa, Japan) and then quantified by an ultraviolet spectrophotometer and agarose electrophoresis. Sequencing libraries were generated using NEBNext®Ultra™ RNA Library Prep Kit for Illumina® (NEB, USA) following the manufacturer's recommendations and index codes were added to attribute sequences to each sample. The clustering of the index-coded samples was performed on a cBot Cluster Generation System using TruSeq PE Cluster Kit v3-cBot-HS (Illumia) according to the manufacturer's instructions. The library preparations were sequenced on an Illumina Hiseq Xten platform. The adaptor sequences and low-quality sequence reads were removed from the data sets. Raw sequences were transformed into clean reads after data processing. These clean reads were then mapped to the wheat reference genome sequence (RefSeq v1.0). Only reads with a perfect match or one mismatch were further analyzed and annotated based on the reference genome using Tophat2 tools software. Gene expression levels were estimated by fragments per kilobase of transcript per million fragments mapped (FPKM). Differential expression analysis of two groups was performed using the DESeq2 R package (1.10.1). false discovery rate (FDR) < 0.01 and fold change (FC) >2 were set as the threshold for significantly differential expression. Gene function was annotated based on the Nr (NCBI non-redundant protein sequences), Nt (NCBI non-redundant nucleotide sequences), Pfam (Protein family), KOG/COG (Clusters of Orthologous Groups of proteins), Swiss-Prot, KO (KEGG Ortholog database), and GO (Gene Ontology). We used KOBAS [25] software to test the statistical enrichment of differential expression genes in KEGG pathways. Transcription factors (TFs) and protein kinases (PKs) were identified and grouped by the iTAK database [26].

## qRT-PCR

A total of 14 pairs of gene-specific primers (S2 Table) were designed to produce amplicons for validating the RNA-seq data. Quantitative reverse-transcription PCR (qRT-PCR) was performed on a LightCycler480 instrument (Rotkreuz, Switzerland) using SYBR Green qPCR kits (Roche) according to the instructions. Relative gene expression levels were calculated using the $2^{-\Delta\Delta Ct}$ method. Expression levels were quantified by normalization against GAPDH. All assays for each gene were performed in triplicate synchronously under identical conditions.

## Statistical analysis

All data were presented as the means ± standard error (SE) of at least three replicates. Statistical analysis (analysis of variance; ANOVA) was performed using SPSS software version 19.0 test for significant differences between different treatment groups. The mean values of each treatment group were compared using Duncan's test at $P < 0.05$.

The ANOVA results of whole-plant bioassay data showed no significant difference between assay repetitions. Then, the repeated assay results were averaged. 50% plant height inhibition ($GR_{50}$) was predicted by 4-parametic log-logistic model analysis using SigmaPlot software (v.12.0) [2].

## Results

### Bioassay

To test the effect of mefenpyr-diethyl to the dose response to mesosulfron-methyl of wheat and Tausch's goatgrass, the $GR_{50}$ was assessed. The results are shown in Table 2 and S1 Fig. It can be seen that the application of mefenpyr-diethyl significantly decreased the sensitivity of both wheat and Tausch's goatgrass to mesosulfuron-methyl. Foliar spray of mefenpyr-diethyl increased the $GR_{50}$ of Tausch's goatgrass and wheat by 7.81 and 7.01 times, respectively. When wheat seeds were dressed with mefenpyr-diethyl, the $GR_{50}$ increased 21.80 times. The doses of mefenpyr-diethyl used for seed dressing were tested before the dose response assay. The results showed that the highest dose without significant growth inhibition was 2 g per kg seed (S1 Table).

### Dissipation of mesosulfuron-methyl

The dissipation curve of mesosulfuron-methyl in Tausch's goatgrass and wheat under different treatments are shown in Fig 1. Dissipation of mesosulfuron-methyl for both Tausch's goatgrass and wheat was fitted reasonably well with a first-order kinetic model, with $R^2$ ranging from 0.9100 to 0.9898. The application of mefenpyr-diethyl by spraying accelerated the dissipation of mesosulfuron-methyl in both Tausch's goatgrass and wheat, since the half-life of MmMdsp was 24.06% and 20.00% shorter than that of Mmsp respectively. Furthermore, the half-life of MmMdsd in wheat was 46.87% shorter than that of Mmsp (Table 3).

### ALS activity, CYP450 content, and GST content assay

To compare the responses of wheat to mesosulfuron-methyl with or without seed dressing, the ALS activity and the content of CYP450 and GST were determined. The results were shown as inhibition rate or relative ratio between Mm and CK or MmMd and Md (Fig 2). The inhibition rate of ALS activity decreased with the time increase. Seed dressing with mefenpyr-diethyl significantly decreased the inhibition rate of ALS activity at all the time points (Fig 2A). The relative ratio of CYP450 and GST content in MmMd/Md were significantly higher than that in Mm/CK at every sampling time. The relative ratio of CYP450 content in MmMd/Md was always below 1.0, while that in Mm/CK was above 1.0 at 1 DAT and decreased thereafter (Fig 2B). The relative ratio of GST content decreased to below 1.0 at 5 and 7 DAT in Mm/CK and MmMd/Md, respectively (Fig 2C).

### Illumina sequencing and PCA plotting

12 RNA libraries constructed from wheat were sequenced, and the raw Illumina sequence reads have been deposited in the NCBI Sequence Read Archive (SRA) database with accession

**Table 2. The response of Tausch's goatgrass and wheat to mesosulfuron-methyl and mefenpyr-diethyl[*].**

| Treatments | $GR_{50}$ (g ai ha$^{-1}$)[#] | | Ratio of $GR_{50}$ | |
|---|---|---|---|---|
| | Tausch's goatgrass | Wheat | Tausch's goatgrass | Wheat |
| mesosulfuron-methyl | 0.32 ± 0.03 [b] | 4.63 ± 0.15 [c] | 1.00 | 1.00 |
| mesosulfuron-methyl + mefenpyr-diethyl (spray) | 2.82 ± 0.15 [a] | 37.09 ± 0.49 [b] | 8.81 | 8.01 |
| mesosulfuron-methyl+ mefenpyr-diethyl (seed dressing) | NA [&] | 105.58 ± 1.44 [a] | NA [&] | 22.80 |

[*] The letters a, b, c in the same column indicate that $GR_{50}$ with different letters are significantly different at the P = 0.05 significance level.

[#] $GR_{50}$: The herbicide dose causing a 50% plant height inhibition. Each value represents the mean ± standard error.

[&] NA: Not available.

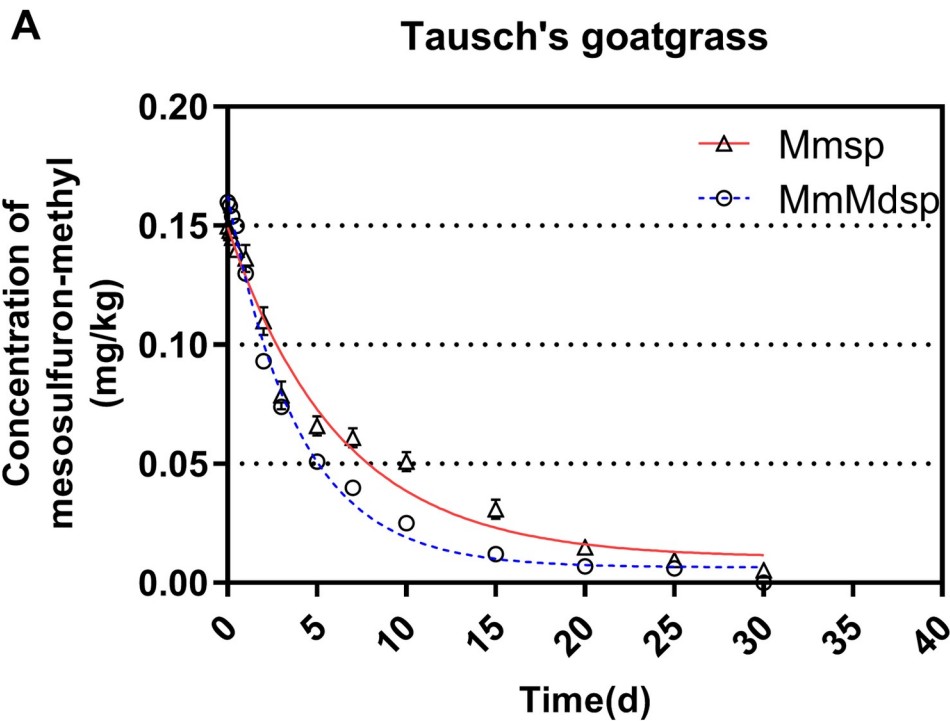

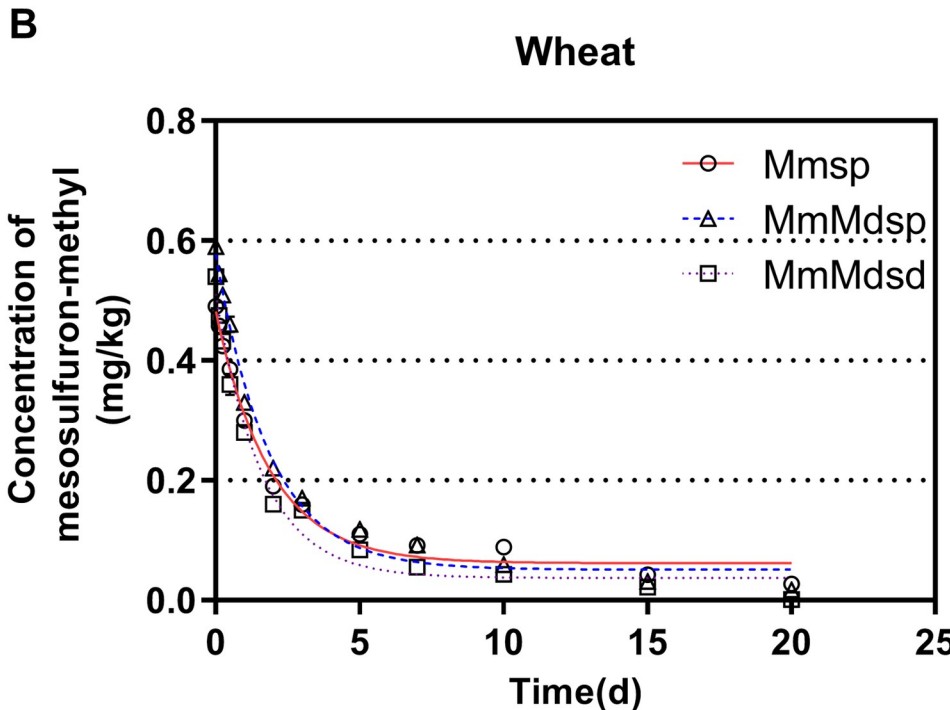

**Fig 1.** Dissipation of mesosulfuron-methyl in Tausch's goatgrass (A) and wheat (B) according to the exponential decay equation. The line represents the first-order regression equation. Data points are the means of three replications, with bars indicating the standard error of the mean. Mmsp: Plants treated with mesosulfuron-methyl by spraying, MmMdsp: Plants treated with mesosulfuron-methyl and mefenpyr-diethyl by spraying, MmMdsd: Plants treated with Mesosulfuron-methyl by spaying and pretreated with mefenpyr-diethyl by seed dressing.

**Table 3. Regression equation, correlation coefficient and half-life of mesosulfuron-methyl in Tausch's goatgrass and wheat under different treatments.**

| Treatments | Regression equation | | Correlation coefficient ($R^2$) | | Half-life (d)[a] | |
|---|---|---|---|---|---|---|
| | Tausch's goatgrass | Wheat | Tausch's goatgrass | Wheat | Tausch's goatgrass | Wheat |
| **mesosulfuron-methyl** | $C = 0.1392\,e^{-0.1083t}$ | $C = 0.3426\,e^{-0.1400t}$ | 0.9898 | 0.9100 | 6.40 | 4.95 |
| **mesosulfuron-methyl + mefenpyr-diethyl (spray)** | $C = 0.1335\,e^{-0.1426t}$ | $C = 0.4099\,e^{-0.1749t}$ | 0.9632 | 0.9423 | 4.86 | 3.96 |
| **mesosulfuron-methyl + mefenpyr-diethyl (seed dressing)** | NA[b] | $C = 0.4125\,e^{-0.2638t}$ | NA | 0.9379 | NA | 2.63 |

[a] Half-life of mesosulfuron-methyl dissipation.

[b] NA: Not available.

number SRP263381. The expression levels (the FPKM values) of all the genes from each sample were used to generate the PCA plot. As can be seen from Fig 3A, the samples primarily clustered into two groups. Samples from Mm were obviously divided from the other three groups.

Differentially expressed genes (DEGs) between groups were identified using the standard of false discovery rate <0.01 and fold change >2. The DEGs in CK vs. Md, CK vs. Mm, and Md vs. MmMd were 1,153, 6,719 and 860, respectively. The up-regulated DEGs were close to that of down-regulated in all the three comparisons (Fig 3B). Venn diagram showed that there were 28 DEGs among the three groups in common (Fig 3C). The DEGs were then subjected to COG database, the result was shown in Fig 4A. Except the DEGs that could not be annotated accurately (general function predicted only), 'posttranscriptional modification, protein turnover, chaperones' and 'secondary metabolites biosynthesis, transport and metabolism', 'transcription' and 'signal transduction mechanisms', 'signal transduction mechanisms' and 'secondary metabolites biosynthesis, transport and metabolism' were the top2 categories in CK vs. Md, CK vs. Mm, and Md vs. MmMd, respectively. In accordance with the COG results, KEGG enrichment showed that DEGs were enriched in glutathione metabolism pathway, lipid metabolites related pathways such as 'alpha-linolenic acid metabolism', 'linoleic acid metabolism', and phenylpropanoid biosynthesis' in CK vs. Md (Fig 4B). Many pathways including 'starch and sucrose metabolism', 'phenylalanine metabolism', and 'phenylpropanoid biosynthesis' were significantly enriched in CK vs. Mm (Fig 4C). Only two pathways, 'starch and sucrose metabolism' and 'sulfur metabolism', were significantly enriched in Md vs. MmMd (Fig 4D).

## Expression of herbicide detoxification related genes

The expression of detoxification related genes, such as *CYP450*, *GST*, *UGTs* (UDP glucuronosyltransferases) and *ABCs* (ATP-binding cassette transporters), are shown in Fig 5. The vast majority of DEGs of detoxification related genes in CK vs. Md were up-regulated, which indicated that seed dressing with mefenpyr-diethyl could enhance the herbicide detoxification of wheat (Fig 5A). The same as that in CK vs. Mm, DEGs in Md vs. MmMd were induced by mesosulfuron-methyl. The only difference between the two comparisons was whether the wheat had been treated with mefenpyr-diethyl by seed dressing. However, the number of DEGs in Md vs. MmMd was much fewer than that in CK vs. Mm. This implied that seed dressing with mefenpyr-diethyl could remarkably reduce the influence of mesosulfuron-methyl to detoxification related genes' expression in wheat (Fig 5B and 5C).

## Transcription factors and protein kinase identification

To identify the transcription factors (TFs) and protein kinase (PKs), DEGs were searched against iTAK database. A total of 749 DEGs were identified, of which 429 were TFs and 320

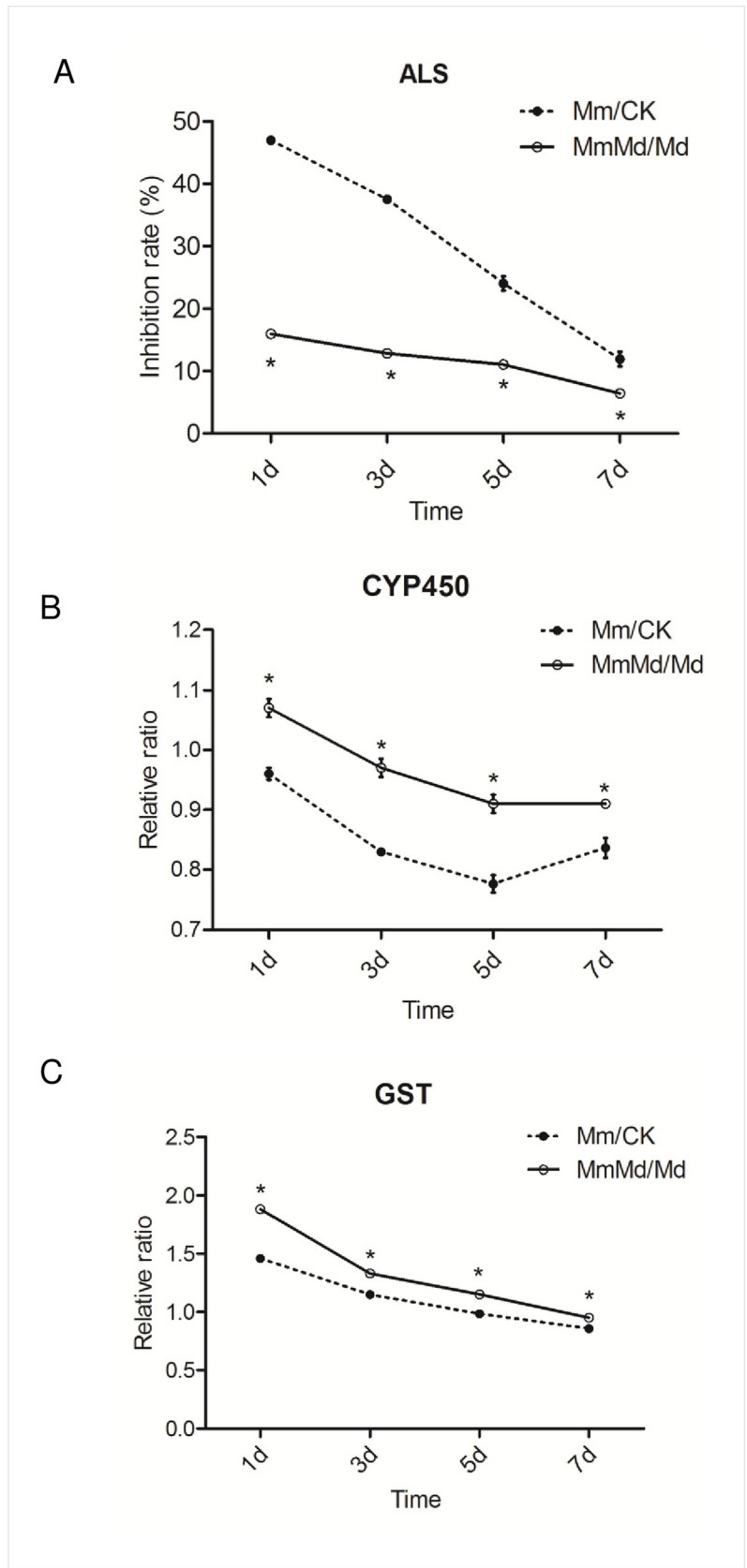

**Fig 2.** Effect of mesosulfuron-methyl and/or mefenpyr-diethyl on ALS activity (A), and CYP450 (B) and GST(C) content in wheat. Error bars indicate standard errors. *Significantly different at the $P < 0.05$ level compared to Mm/CK.

were PKs (S3 Table). As can be seen from Fig 6, DEGs belonging to *AP2/ERF-ERF*, *bHLH*, *NAC*, and *MYB* families were largely involved in the responses to mefenpyr-diethyl and meso-sulfuron-methyl. Interestingly, the expression of *AP2/ERF-ERF* and *MYB* was up-regulated in CK vs. Md, mixed regulated in CK vs. Mm and Md vs. MmMd. The expression of *bHLH* was mixed regulated in CK vs. Md and CK vs. Mm, but down-regulated in Md vs. MmMd. And the expression of *NAC* was up-regulated in CK vs. Md, mixed regulated in CK vs. Mm and down-regulated in Md vs. MmMd. *RLK-Pelle DLSV* was the most enriched PKs in all comparisons, and the expression of which was mixed regulated. We further analyzed the correlations

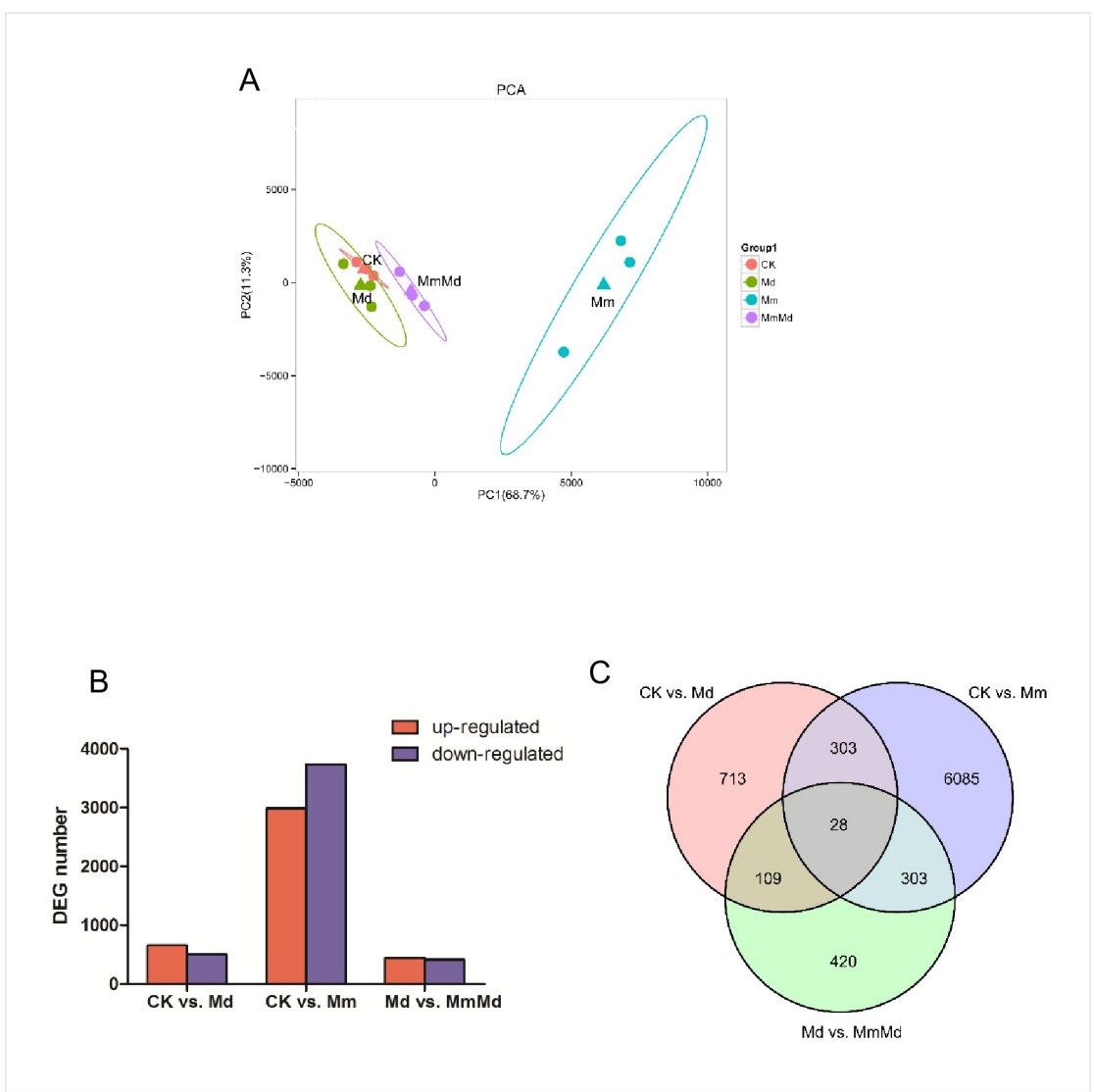

**Fig 3. PCA plot and differentially expressed genes (DEGs).** PCA plot based on the FPKM of all expressed genes (A). Numbers of up- and down-regulated genes in CK vs. Md, CK vs. Mm, and Md vs. MmMd (B). Venn diagram of differentially expressed genes (DEGs) (C).

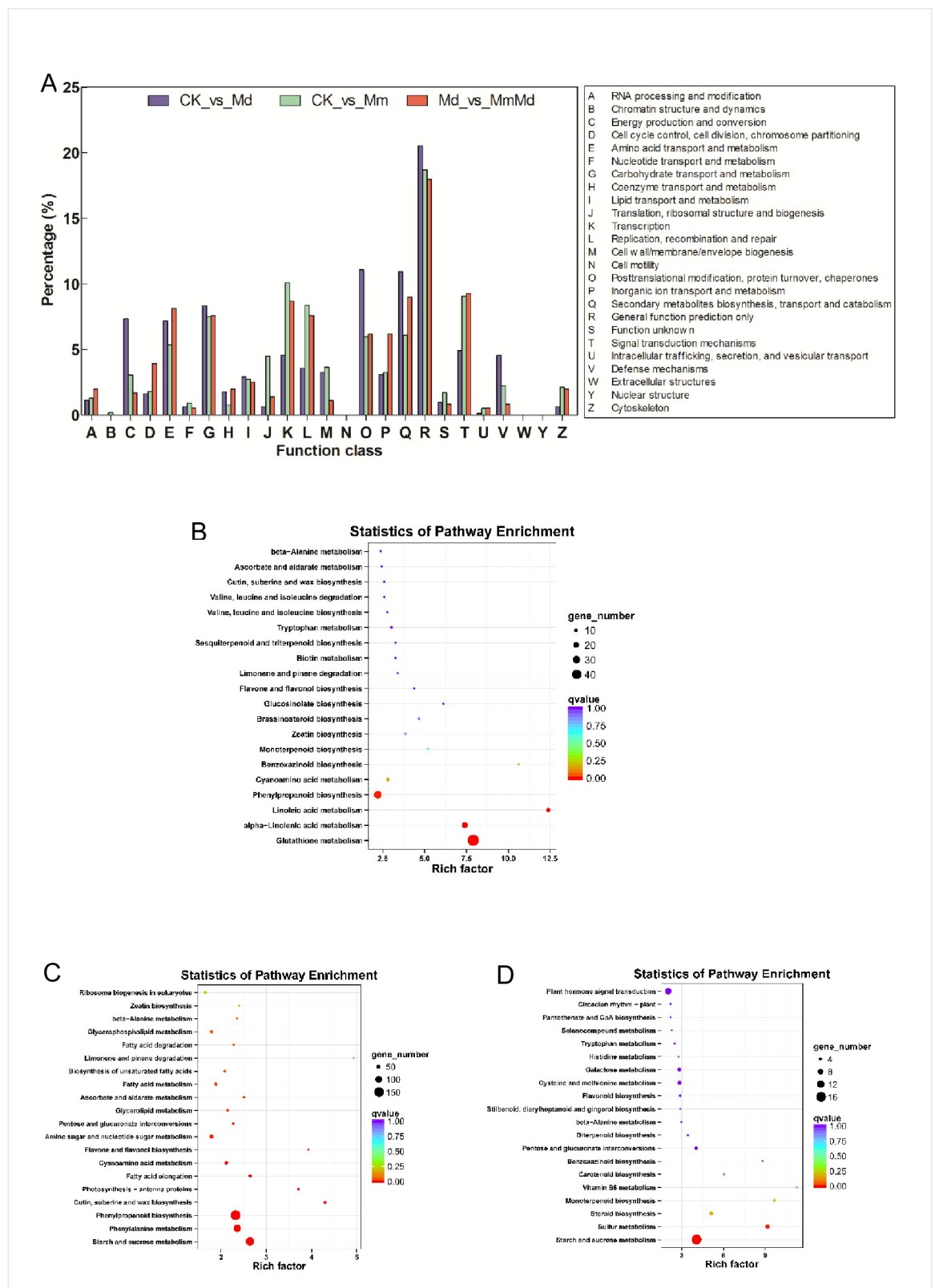

**Fig 4. COG and KEGG enrichment analysis.** COG annotation of differentially expressed genes (DEGs) between different groups (A). KEGG enrichment of DEGs in CK vs. Md (B), CK vs. Mm (C), and Md vs. MmMd (D).

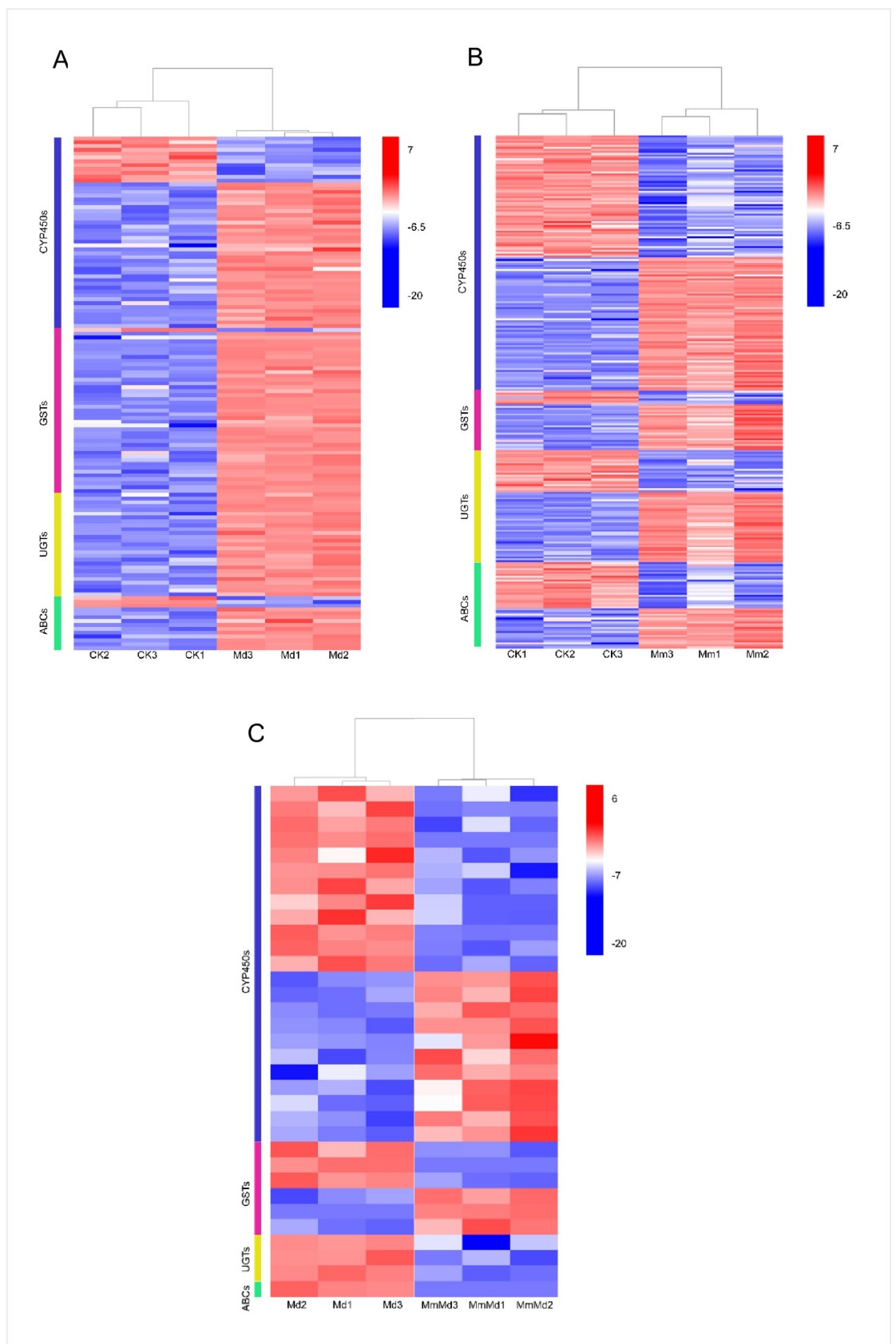

**Fig 5. Expression of detoxification-related genes in wheat.** Heatmap of expression changes in differentially expressed genes (DEGs) in CK vs. Md (A), CK vs. Mm (B), and Md vs. MmMd (C).

of some of the transcription factors with ALS activity and GST and CYP450 content. Several transcription factors, including *DREB*, *TGA*, *WRKY*, and *bHLH* were positively or negatively correlated with ALS activity or GST and CYP450 content. However, only *DREB (TraesCS7D01G127600)*, which belongs to *AP2/ERF-ERF* family, was significantly correlated (spearman r = 0.9429, P = 0.0167) with GST content among the selected transcription factors.

## qRT-PCR

To validate the reliability of the RNA-seq data, 13 genes were selected to perform qRT-PCR in all samples. The relative expression levels of these genes were similar to the expression profiles determined from the respective RNA-seq data and the correlation ($R^2$) between the qRT-PCR and RNA-seq results was above 0.94 (Fig 7A), suggesting the accuracy of the RNA-seq data. Furthermore, we detected the expression level of two ALS genes, *ALS1* and *ALS2*, at 0, 1/2, 1, 3, 5, 7 DAT (Fig 7B and 7C). The expression levels of both *ALS1* and *ALS2* were significantly upregulated at 1 and 3 DAT in Mm and MmMd. *ALS1* in MmMd was also significantly upregulated at 1/2 DAT.

## Discussion

Herbicide safeners have been generally considered to selectively protect crops from herbicide damage with little or no effect on target weeds [27–30]. However, studies found that certain safeners could improve the tolerance of weeds. For example, mefenpyr-diethyl and fenchlorazole ethyl enhanced the tolerance to fenoxaprop ethyl of black-grass (*Alopecurus myosuroides* Huds.) [10]. Duhoux *et al.* demonstrated a reduction in the sensitivity of rye-grass (*Lolium* sp.) to ALS inhibiting herbicides pyroxsulam and iodosulfuron + mesosulfuron induced by safeners cloquintocet-mexyl and mefenpyr-diethyl respectively [12]. Furthermore, accelerated herbicide metabolism is one of the most important safener mechanisms [13,27,28] and the metabolic pathways are strikingly similar to those involved in non-target-site-based resistance (NTSR) in weeds [31,32]. An increasing number of studies raised the issue of a possible role of safeners on NTSR evolution in weeds [12,33,34]. Thus, it is of great agronomic value to make clear the effects of safeners on the tolerance of weeds and to establish the technology which delivers safeners exclusively to the crop.

The first question in this study sought to determine was whether spraying of mefenpyr-diethyl would increase the tolerance of Tausch's goatgrass to mesosulfuron-methyl. Similar to the previous studies [10,12], the results in our study indicated that spraying of mefenpyr-diethyl increased the $GR_{50}$ of mesosulfuron-methyl to Tausch's goatgrass by 7.81 times and reduced the half-life time of mesosulfuron-methyl by 24.06%, while smaller numbers were obtained from wheat as 7.01 and 20.00% respectively (Tables 2 and 3). The present study therefore tested seed dressing as an alternative method to avoid the protection to Tausch's goatgrass by mefenpyr-diethyl. As expected, seed dressing was proved to be efficient at increasing the tolerance of wheat. Furthermore, the half-life of mesosulfuron-methyl of MmMdsd was shorter than that of Mmsp, suggesting seed dressing accelerated the dissipation process. We therefore suggest more researches to be done on the feasibility of replacing the commonly used spraying method with seed dressing as the field application way of mefenpyr-diethyl in the control of Tausch's goatgrass.

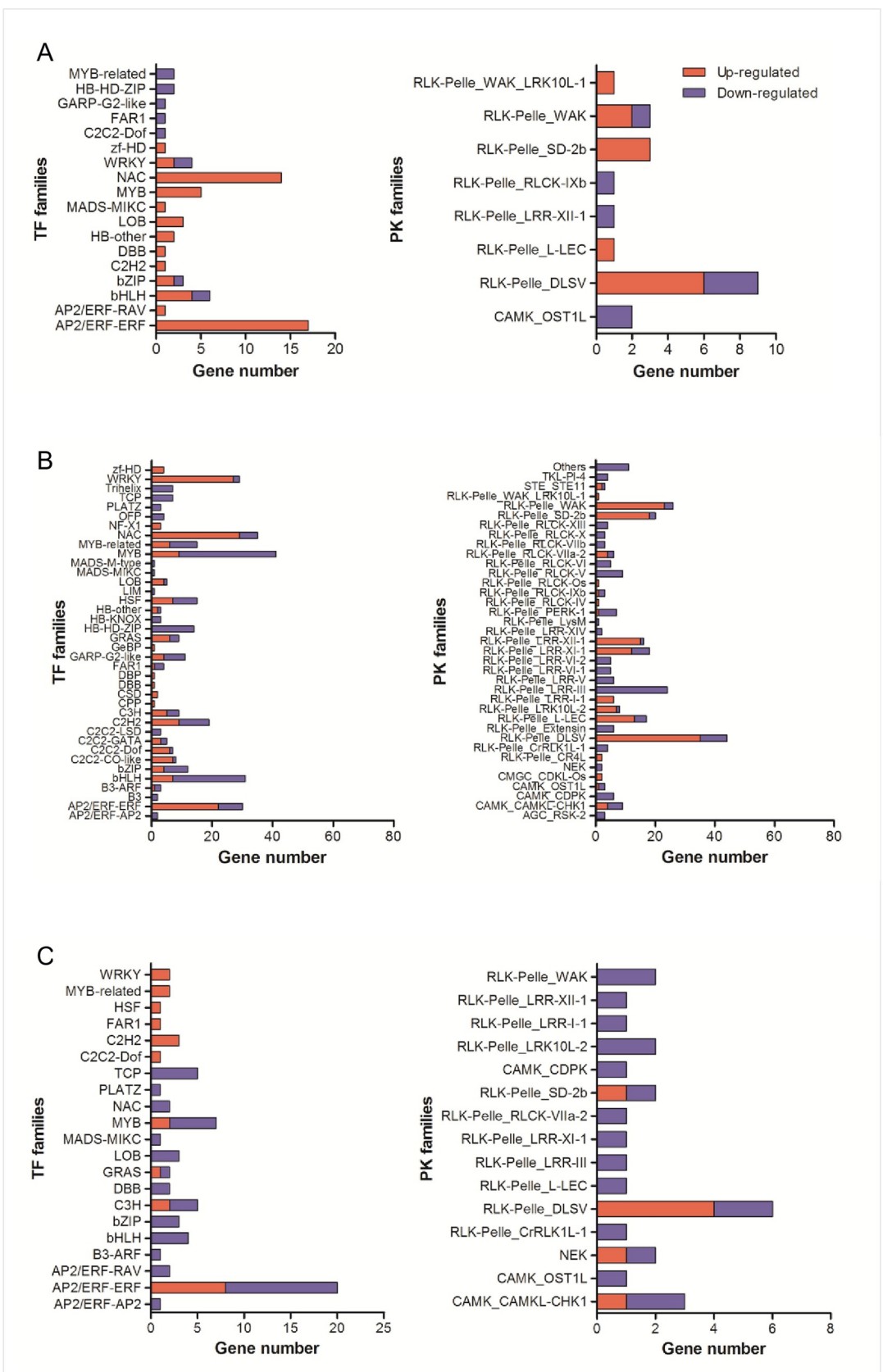

**Fig 6. Transcription factors and protein kinases in wheat.** Numbers of differentially expressed genes (DEGs) annotated as transcription factors (TFs) and protein kinases (PKs) in CK vs. Md (A), CK vs. Mm (B), and Md vs. MmMd (C).

It is noticed that the dissipation rate of mesosulfuron-mehtyl was higher in wheat than in Tausch's goatgrass (Table 2), indicating that mesosulfuron-methyl metabolism was faster in wheat. This finding is in agreement with Köcher *et al.* who demonstrated the selectivity of mesosulfuron-methyl was based on the metabolism difference between wheat and weed [17].

ALS is the target enzyme of mesosulfuron-methyl and its activity change under herbicidal stress can be an index of the tolerance of plants [6]. ALS activity of wheat was significantly inhibited in the presence of mesosulfuron-methyl, but the inhibition was attenuated by mefenpyr-diethyl (Fig 2A). This is in line with previous reports for maize [35–37]. The increase in ALS activity (S2 Fig) may be attributed to the increase of ALS expression (Fig 7B and 7C) and the safener's competition with the herbicide [35].

A growing body of literatures has demonstrated that herbicides have a well-defined and characterized detoxification process in plants [13,32,38]. The process generally include four steps, that are, oxidation and hydrolyses (phase I), conjugations to endogenous molecules (phase II), transport of conjugates (phase III), and processing of conjugates (phase IV) [30]. CYP450 and GST are crucial enzymes involved in phases I and II, respectively. Some studies reported the involvement of CYP450 in herbicide metabolism [39,40]. Increase in CYP450 content was also observed in the current study (S2 Fig). Interestingly, the relative ratio of CYP450 content between MmMd/Md was significantly higher than that of Mm/CK (Fig 2B), suggesting the safener could weaken the inhibition on CYP450 caused by mesosulfuron-methyl. Similar results were observed with GST levels. Consistent with previous studies [41,42], GST was induced by safener at 3 DAT. However, the increase was transient in that GST content decreased at 5 and 7 DAT (S2 Fig). A possible reason may be that we determined the total GST content, whilst GST enzymes include the tau, phi, and lambda classes. Analogously, Andrew *et al.* reported a selective enhancement of GST isoenzymes caused by herbicides and herbicide safeners in soybean [43]. In addition, it could be an evidence that *GST* genes were observed to be mixed regulated in this study. It is noted that, although both CYP450 and GST content were induced by the herbicide, the relative ratio of CYP450 content was < 1 at most time points, while that of GST content was chiefly >1 (Fig 2B and 2C), indicating that the herbicide would depress CYP450 and activate GST under detected rates regardless of the absence or presence of the safener. The expression of *UGTs* and *ABCs*, which are important enzymes in phases II and III respectively, was also induced by mefenpyr-diethyl. Expect for conjunction with glutathione (GSH), the products in phase I may also undergo glycosylation mediated by UGTs. Several reports have demonstrated that safeners enhance the glycosylation of herbicides in protected plants [30,44]. The current study had drawn the same conclusion since all the *UGTs* genes were up-regulated induced by mefenpyr-diethyl (Fig 5A). The GSH- and glycosyl- conjugates are transported into the vacuole of plant cells by transporters, including *ABCs*. However, *ABCs* could have different substrates and transport characteristics [45]. It may partly explain as to why *ABCs* genes were mixed regulated (Fig 5A). Collectively, the presence of mefenpyr-diethyl protected wheat from mesosulfuron-methyl via inducing the expression of metabolic enzymes, thus enhancing herbicide detoxification.

Herbicide-regulated pathways are considered to be involved in general stress responses, while safener-responsive pathways are mostly involved in xenobiotic detoxification [46]. There are several theories for the mechanisms of toxicity induced by ALS inhibitors. These theories involved unusual accumulation of an intermediate, 2-ketobutyrate and/or 2-aminobutyrate [47], depletion of the free branched-chain amino acid pool [48], inhibition of assimilate

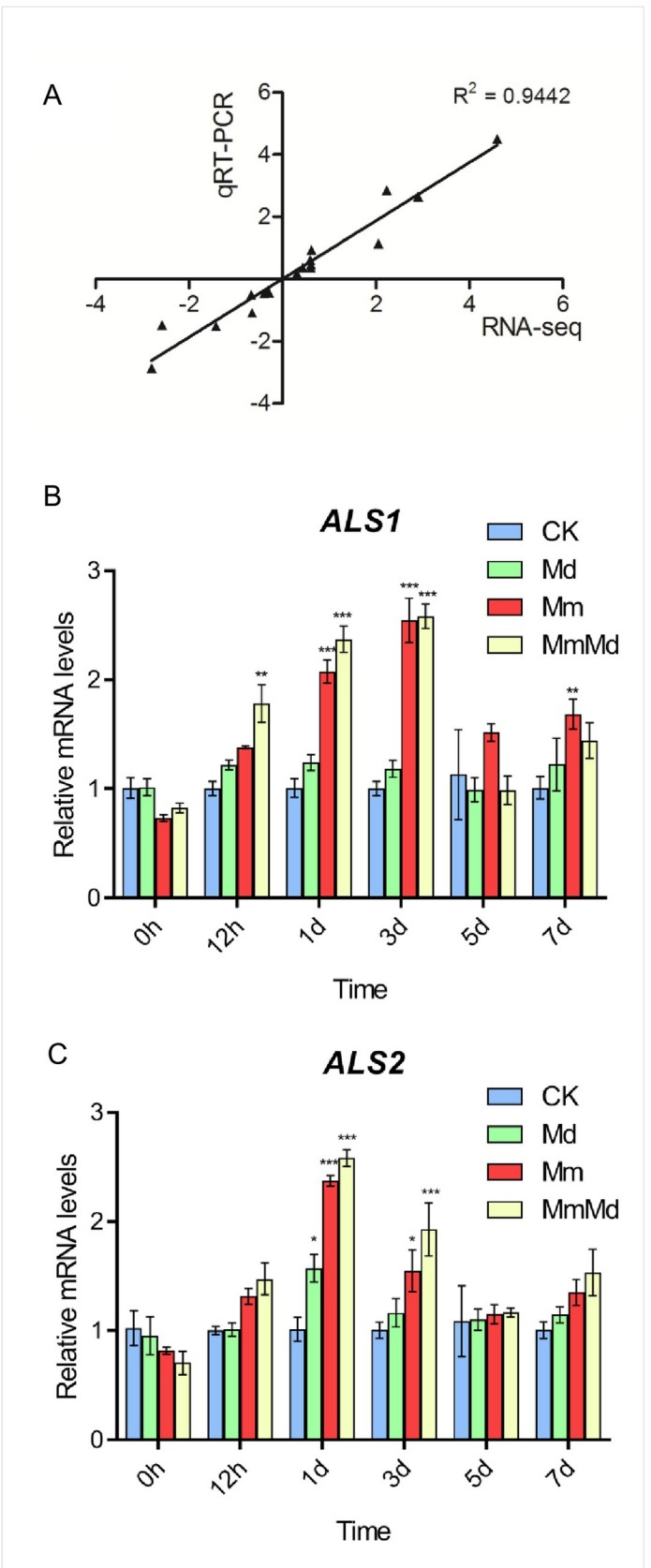

**Fig 7. qRT-PCR validation of RNA-seq data.** Correlation of fold changes determined by RNA-Seq and qRT-PCR data (A). Relative expression levels of *ALS1* (B) and *ALS2* (C).

transport and utilization [49], and fermentative metabolism induction [50], etc. Some other effects were also elicited by sulfonylurea, such as blocks of carbohydrate and lipid metabolism and autophagic cell death [51]. Certain pathways related to the effects mentioned above were enriched in this study. For example, carbohydrate and lipid metabolism was enriched as 'starch and sucrose metabolism' and 'fatty acid metabolism' (Fig 4C). Besides, 'phenylalanine metabolism' and 'phenylpropanoid biosynthesis', which were two less studied pathways elicited by ALS inhibitors, were significantly enriched induced by mesosulfuron-methyl (Fig 4C). These two pathways are actually part of the early responsive pathways, which have also been shown to be induced by other abiotic stresses [52]. Since these two pathways provide precursors for a wide range of important secondary metabolites, their enrichment implicated that mesosulfuron-methyl could possibly influence the secondary metabolism of wheat via 'phenylalanine metabolism' and 'phenylpropanoid biosynthesis' pathways.

At the presence of both mesosulfuron-methyl and mefenpyr-diethyl, it seems that the response was not a simple addition of the two chemicals. Safener was hypothesized to tap into the preexisting pathways for detoxification via a oxylipin-mediated pathway, facilitating the expression of GST [53]. It is found in the current study that lipoxygenase, the key enzyme in the biosynthesis of oxylipins, and GST, as the result of safener alone, were both significantly enriched (S4 Table), supporting the proposed hypothesis. Interestingly, it was documented many transgenic plants overexpressing serine O-acetyltransferase (SAT), the key enzyme in sulfur metabolism pathway, had higher GST activity [54]. Since SAT genes were enriched (S4 Table) and up-regulated (S3 Fig) as the result of safener together with herbicide, it is speculated that certain GST may be triggered by SAT in the presence of both herbicide and safener, whilst GST were triggered by safener alone via the oxylipin-mediated pathway.

Since 'secondary metabolites biosynthesis, transport and metabolism' and 'signal transduction mechanisms' were two of the most enriched COG terms (Fig 4A), we set out to identify TFs and PKs involved in these processes. A number of TFs had been characterized to regulate expression of genes involved in biosynthesis of secondary metabolites and mediating the biotic and abiotic stress responses [55,56]. For instance, TFs belonging to the *MYB* and *bHLH* regulated key enzymes of flavonoid biosynthesis and phenylpropanoid metabolism [57,58]. *NAC* and *AP2/ERFs* are important transcriptional regulators related to plant strategies under conditions of stresses [59,60]. Some *bZIP* transcription factors are critical in safener-mediated detoxification and defense [30]. In the current study, all these TFs were found to alter their expressions, and some had similar expression patterns with ALS activity, GST or CYP450 content, suggesting their participation in the detoxification or stress response processes. Further investigations may focus on *DREB* or certain TFs to formulate their regulatory networks. PKs play a pivotal function in plant signal transduction [61]. *RLK/Pelle* family is the largest class of protein kinase in plants and it is a group of conserved signaling components that regulate growth, development and responses to biotic and abiotic stimuli [62,63]. In this study, most of the enriched PKs also belong to this family (Fig 6).

The safener mefenpyr-diethyl is commonly used together with mesosulfuron-methyl, which is the most important herbicide to control Tausch's goatgrass in wheat fields in China, by foliar spraying. However, this study found out that the spraying of mefenpyr-diethyl could remarkably increase Tausch's goatgrass' tolerance to mesosulfuron-methyl, which might result in the waste of herbicide and environmental pollution. For a possible alternative method, seed dressing may deliver the safener exclusively to the wheat without improving the tolerance of

Tausch's goatgrass. The results have important implications for mefenpyr-diethyl application in the control of Tausch's goatgrass. The herbicide dissipation study, enzymatic analysis and transcriptome data disclosed that the mechanisms of mefenpyr-diethyl used as a safener by seed dressing may involve increase of ALS activity, enhancement of metabolic processes, and other stress responses. In addition, a lot of differentially expressed genes were identified as TFs and PKs, suggesting a complex regulatory system for response to mefenpyr-diethyl, some of which deserve further investigations.

## Supporting information

**S1 Fig.** Dose-response curve of Tausch's goatgrass (A) and wheat (B) treated with different doses of mesosulfuron-methyl with or without mefenpyr-diethyl applied by spraying or seed dressing. Plant height was expressed as a percentage of the untreated control. Each data point represents the mean ± SE of twice-repeated experiments containing three replicates each, and vertical bars represent the standard error. Mmsp: Plants treated with mesosulfuron-methyl by spraying, MmMdsp: Plants treated with mesosulfuron-methyl and mefenpyr-diethyl by spraying, MmMdsd: Plants treated with Mesosulfuron-methyl by spaying and pretreated with mefenpyr-diethyl by seed dressing.
(TIF)

**S2 Fig.** Effect of mesosulfuron-methyl and/or mefenpyr-diethyl on the ALS activity (A), and CYP450 (B) and GST(C) content. Error bars indicate standard errors. *Significantly different at the $P < 0.05$ level compared to CK.
(TIF)

**S3 Fig. Expression changes of sulfur metabolism related genes after mesosulfuron-methyl application in wheat treated by seed dressing with mefenpyr-diethyl.**
(TIF)

**S1 Table. Growth response of wheat treated by seed dressing with different mefenpyr-diethyl application doses.**
(DOCX)

**S2 Table. Primers used for qRT-PCR.**
(DOCX)

**S3 Table. Number of DEGs identified as transcription factors (TFs) and protein kinases (PKs).**
(DOCX)

**S4 Table. Enzymes related to DEGs in the most enriched pathways.**
(DOCX)

## Acknowledgments

The authors would like to thank Chaoxian Zhang, Yaofa Li, Wei Li, Hongyu Chen, Weitang Liu, Tao Zhang for their helpful suggestions on the draft.

## Author Contributions

**Conceptualization:** Libing Yuan, Wenliang Pan.

**Data curation:** Jian Li.

**Formal analysis:** Libing Yuan.

**Funding acquisition:** Libing Yuan.

**Investigation:** Yaling Geng, Hua Wang, Shanshan Song, Zhiying Hun.

**Methodology:** Libing Yuan.

**Project administration:** Libing Yuan.

**Resources:** Yaling Geng, Hua Wang, Shanshan Song, Zhiying Hun.

**Software:** Jian Li.

**Supervision:** Libing Yuan.

**Visualization:** Jian Li.

**Writing – original draft:** Libing Yuan, Guangyuan Ma, Jian Li.

**Writing – review & editing:** Xiaomin Liu, Wenliang Pan.

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
