## [Decision Letter · Decision Letter 0]

7 Jul 2021

PONE-D-21-06771

Seed dressing with mefenpyr-diethyl as a safener for mesosulfuron-methyl application in wheat:the evaluation and mechanisms

PLOS ONE

Dear Dr. Yuan,

Thank you for submitting your manuscript to PLOS ONE. After careful consideration, we feel that it has merit but does not fully meet PLOS ONE’s publication criteria as it currently stands. Therefore, we invite you to submit a revised version of the manuscript that addresses the points raised during the review process.

We look forward to receiving your revised manuscript.

Kind regards,

Vijay Gahlaut, Ph.D.

Academic Editor

PLOS ONE

Journal Requirements:

2. We note that you are reporting an analysis of a microarray, next-generation sequencing, or deep sequencing data set. PLOS requires that authors comply with field-specific standards for preparation, recording, and deposition of data in repositories appropriate to their field. Please upload these data to a stable, public repository (such as ArrayExpress, Gene Expression Omnibus (GEO), DNA Data Bank of Japan (DDBJ), NCBI GenBank, NCBI Sequence Read Archive, or EMBL Nucleotide Sequence Database (ENA)). In your revised cover letter, please provide the relevant accession numbers that may be used to access these data. For a full list of recommended repositories, see http://journals.plos.org/plosone/s/data-availability#loc-omics or http://journals.plos.org/plosone/s/data-availability#loc-sequencing.

Reviewers' comments:

Reviewer's Responses to Questions

**Comments to the Author**

1. Is the manuscript technically sound, and do the data support the conclusions?

Reviewer #1: Yes

Reviewer #2: Yes

2. Has the statistical analysis been performed appropriately and rigorously? 

Reviewer #1: Yes

Reviewer #2: Yes

3. Have the authors made all data underlying the findings in their manuscript fully available?

Reviewer #1: Yes

Reviewer #2: Yes

4. Is the manuscript presented in an intelligible fashion and written in standard English?

Reviewer #1: Yes

Reviewer #2: No

5. Review Comments to the Author

Reviewer #1: The application of mefenpyr-diethyl as seed dresser in wheat is a novel approach and a quite promising technique to mitigate the adverse effects of mesosulfuron-methyl and in general of many herbicides to the crop. The subject of the study is interesting with high scientific and practical importance.

The introduction is in accordance with the subject and correctly presented.

Methodology of the study was clearly presented and was appropriate to the proposed objectives, allowing the repetition of the experiments and the protocols. The statistical analysis was clear.

The obtained results have been analyzed and interpreted in accordance with the current methodology. The thorough analysis of the agronomic traits (GR50), dissipation, enzymic and transcriptomic assays resulted to a significant outcome.

The discussion is appropriate, in the context of the results, and was conducted compared to other studies in the field. Authors are strongly encouraged to cite more articles about the agronomic value of seed dressing with safener, in order to increase the impact of their research.

The scientific literature, to which the reporting was made, is recent and representative in the field.

However, authors are kindly asked to answer: (1) why they applied mesosulfuron-methyl with/without mefenpyr-diethyl at 1-leaf stage of the wheat and the weed. It is quite uncommon in the field scale to spray herbicides in such early growth stage, (2) why they used plant height instead of plant fresh/dry weight in the dose-response assays, (3) why they mention weed resistance in the abstract without linking their results about the metabolic changes in the plants with existing literature about ALS resistance mechanisms.

Moreover, the review of the article revealed some minor issues, which are noted below.

Keywords: safener

Line 23: It is probably useful to refer to the seed dressing of wheat by writing ..the effect of wheat seed dressing..

Lines 36-37: Include the binomial names of the species in italics

Line 37: “It could cause 50-80% yield loss of production in wheat producing regions”. Which is the density that may lead to such yield loss?

Line 38: “were”. I believe that remain poor until today, so please edit appropriately.

Line 49: increase

Line 53: involves

Lines 54-56: Please rephrase the sentence “However..safeners”.

Lines 76-77: “No suspected resistant or susceptible to mesosulfuron-methyl of Taush’s goatgrass was found to the land owner”. Authors mention both lack of resistance and sensitivity to mesosulfuron-methyl. Which is correct? If the Tausch's goatgrass biotype is susceptible, then why is resistance mentioned? Which is the history of herbicide applications in the field? Was mesosulfuron-methyl consecutively applied in wheat every year?

Line 86: mesosulfuron

Lines 151/154: ANOVA

Line 171: Authors mention here that GR50 refers to the 50% growth inhibition of plants, while previously it is noted as plant height inhibition. Which is correct?

Line 290-291: You mean probably mesosulfuron-methyl and not mesulfuron-methyl

Line 296: mefenpyr-diethyl

Reviewer #2: Mesosulfuron-methyl is a specific herbicide to control Tausch’s goatgrass which is the most troublesome weeds in wheat fields in China. Without safener, mesosulfuron-methyl will cause certain damage to wheat. This manuscript reported the effect and mechanism of mefenpyr-diethyl as a safener for mesosulfuron-methyl application, and provided an alternative approach to reduce the crop injury and enhance the herbicide efficacy by seed dressing. In general, this manuscript is well organized and the finding is of great importance for weed management in wheat fields.

To improve the quality and readability, some suggestions or comments are listed as the following：

Line 19: need to add latin name for plant like wheat or Tausch’s goatgrass at the first time it is used in the abstract or the body of the paper.

Line 20: suggest change “varieties” to “cultivars”.

Line 24: suggest change “uncovered” to “disclosed”.

Line 25: change “enzymic” to “enzymatic”.

Line 28: suggest change “RLK-Pelle DLSV” to “RLK-Pelle_DLSV”.

Line 36-37: Plant latin name should be italicized.

Line 48: change “no reports” to “no report”.

Line 49: change “increas” to “increase”, need to check the grammar and spelling throughout the manuscript.

Line 51: delete “and increase the risk of phytotoxicity”.

Line 61: delete “herbicide”.

Line 93-95: in table 1, need to check the dose of the treatment.

Line 112-122: Why did the authors not to study the mechanisms of spaying both mesosulfuron-methyl and mefenpyr-diethyl for comparison and concentrate only on seed dressing?

Line 151 and after: change “ANVOA” to “ANOVA”.

Line 167: in table 2, the ratio of GR50 should be listed in the table.

Line 205-237: For transcriptome analysis, common sequence results and gene annotations maybe omitted, the authors should focus on DEGs and their dynamics related to certain pathways like stress responses, transcription factors and protein kinases which were induced by mefenpyr-diethyl.

Line 239-240: For consistency, gene name like UGTs and ABCs should change position with their definitions.

Line 278 and after: the discussion section should be concise and informative, for practical use of seed dressing in weed management, the authors need to analyze the common or difference between this study and other previous studies, based on the evaluation of effects, risks or potential usage in current advances, the authors could make an objective and rigorous conclusion.

Line 293: change “in” to “between”.

6. PLOS authors have the option to publish the peer review history of their article (what does this mean?). If published, this will include your full peer review and any attached files.

Reviewer #1: No

Reviewer #2: No

---

## [Author Response · Author response to Decision Letter 0]

17 Jul 2021

Dear Editor and Reviewers:

Thank you for your letter and for the reviewers’ comments concerning our manuscript entitled “Seed dressing with mefenpyr-diethyl as a safener for mesosulfuron-methyl application in wheat：the evaluation and mechanisms” (PONE-D-21-06771). Those comments are all valuable and very helpful for revising and improving our paper, as well as of important guiding significance to our researches. We have studied the comments carefully and made corrections which we hope meet with approval. Revised portions are marked under track changes mode in MS. The main corrections in the paper and the responds to the reviewer’s comments are as following:

NOTE: the red words are comments or questions from the editor or reviewers, and the black words are our replies.

Reviewer #1: 

1. The application of mefenpyr-diethyl as seed dresser in wheat is a novel approach and a quite promising technique to mitigate the adverse effects of mesosulfuron-methyl and in general of many herbicides to the crop. The subject of the study is interesting with high scientific and practical importance. The introduction is in accordance with the subject and correctly presented. Methodology of the study was clearly presented and was appropriate to the proposed objectives, allowing the repetition of the experiments and the protocols. The statistical analysis was clear. The obtained results have been analyzed and interpreted in accordance with the current methodology. The thorough analysis of the agronomic traits (GR50), dissipation, enzymic and transcriptomic assays resulted to a significant outcome. The discussion is appropriate, in the context of the results, and was conducted compared to other studies in the field.

Thank you for your time and effort put in reviewing our manuscript. We deeply appreciate your consideration of our manuscript.

2. Authors are strongly encouraged to cite more articles about the agronomic value of seed dressing with safener, in order to increase the impact of their research.

Thank you for your constructive suggestion. We have cited 11 articles in the first paragraph of the discussion to emphasize the agronomic value of seed dressing, which can be seen in the revised manuscript.

3. Why they applied mesosulfuron-methyl with/without mefenpyr-diethyl at 1-leaf stage of the wheat and the weed. It is quite uncommon in the field scale to spray herbicides in such early growth stage?

Thank you for your question. As is pointed out, in field scale, mesosulfuron-methyl is always sprayed at about 3-leaf stage of wheat and weed. At the beginning of the bioassay in our study, mesosulfuron-methyl with/without mefenpyr-diethyl was applied at 3-leaf stage, too. However, because wheat itself was partly tolerant to mesosulfuron-methyl, the inhibition rate of the herbicide on wheat was less than 50% even at the highest dosage that we could formulate from the mesosulfuron-methyl technical material. The inhibition rate was too small to determine the dose-response curve accurately. Since wheat was much more susceptive to mesosulfuron-methyl at 1-leaf stage than that at 3-leaf stage, the herbicide was applied to wheat at 1-leaf stage to determine the dose-response curve. For consistency, mesosulfuron-methyl was also applied to Tausch’s goatgrass at 1-leaf stage.

4. Why they used plant height instead of plant fresh/dry weight in the dose-response assays?

Xie (2004) compared four indexes including plant height, root length, fresh weight and dry weight to screen the most desirable index for the dose-response assay of mesosulfuron-methyl to wheat. Finally, plant height was chosen, since it was more sensitive and more dependable. Some other researches (Gao et al., 2011; Fritz et al., 2009; Senarathne et al., 2009; Dias et al., 2021) also used plant height in the dose-response assays, indicating that plant height may be a desirable index. We therefore used plant height herein.

5. Why they mention weed resistance in the abstract without linking their results about the metabolic changes in the plants with existing literature about ALS resistance mechanisms?

Thank you for your reminding. Based on the existing literatures and the results of our study, we have supplemented the discussion on the safeners’ effect on the weed resistance evolution in the first paragraph of the discussion.

6. Keywords: safener.

Thank you for your reminding. We feel sorry for our carelessness. We have corrected it and checked our spelling and grammar throughout the manuscript. 

7. Line 23: It is probably useful to refer to the seed dressing of wheat by writing ..the effect of wheat seed dressing. 

Thank you for your suggestion. The sentence has been rewritten.

8. Lines 36-37: Include the binomial names of the species in italics

Thank you for your reminding. The binomial names in the manuscript have all been changed to italic font now.

9. Line 37: “It could cause 50-80% yield loss of production in wheat producing regions”. Which is the density that may lead to such yield loss?

As can be seen in the manuscript, this sentence is cited from another article, where the density causing such yield loss is not described. We further searched some other literature that concerned this information and found some detail descriptions. Based on the study (Zhang et al., 2007), we speculate the density that causes such yield loss may be over 457 inflorescences per m2.

10. Line 38: “were”. I believe that remain poor until today, so please edit appropriately.

Thank you for your reminding. The word “were” has been changed to “are”.

11. Line 49: increase.

Thank you for your reminding. It has been corrected now.

12. Line 53: involves

Thank you for your reminding. It has been corrected now.

13. Lines 54-56: Please rephrase the sentence “However..safeners”

Thank you for your reminding. The sentence has been re-organized now. 

14. Lines 76-77: “No suspected resistant or susceptible to mesosulfuron-methyl of Taush’s goatgrass was found to the land owner”. Authors mention both lack of resistance and sensitivity to mesosulfuron-methyl. Which is correct? If the Tausch's goatgrass biotype is susceptible, then why is resistance mentioned? Which is the history of herbicide applications in the field? Was mesosulfuron-methyl consecutively applied in wheat every year?

The sentence has been reworded as “No suspected resistance to mesosulfuron-methyl of the Taush’s goatgrass was found according to the land owner”. Mesosulfuron-methyl was applied only in the former year instead of consecutively applied in wheat field. The description of this history has been added to the manuscript.

15. Line 86: mesosulfuron

Thank you for your reminding. It has been corrected now.

16. Lines 151/154: ANOVA.

Thank you for your reminding. It has been corrected now. 

17. Line 171: Authors mention here that GR50 refers to the 50% growth inhibition of plants, while previously it is noted as plant height inhibition. Which is correct?

Thank you for your reminding. The term GR50 refers to the 50% plant height inhibition of plants and the definition has been corrected herein

18. Line 290-291: You mean probably mesosulfuron-methyl and not mesulfuron-methyl.

Thank you for your reminding. It has been corrected now. 

19. Line 296: mefenpyr-diethyl.

Thank you for your reminding. It has been corrected now.

Reviewer #2:

1. Mesosulfuron-methyl is a specific herbicide to control Tausch’s goatgrass which is the most troublesome weeds in wheat fields in China. Without safener, mesosulfuron-methyl will cause certain damage to wheat. This manuscript reported the effect and mechanism of mefenpyr-diethyl as a safener for mesosulfuron-methyl application, and provided an alternative approach to reduce the crop injury and enhance the herbicide efficacy by seed dressing. In general, this manuscript is well organized and the finding is of great importance for weed management in wheat fields.

Thank you for your appreciation and careful revision on our manuscript. We have reorganized our manuscript according to your and the other reviewer’s suggestions. We made some necessary corrections and tried our best to make our manuscript more readable. We hope the corrections could meet your approval.

2. Line 19: need to add latin name for plant like wheat or Tausch’s goatgrass at the first time it is used in the abstract or the body of the paper.

Thank you for your reminding. The latin names of wheat and Tausch’s goatgrass have been added at its first appearance in the abstract and the body of the paper.

3. Line 20: suggest change “varieties” to “cultivars”.

Thank you for your suggestion. The word “varieties” has been changed to “cultivars”.

4. Line 24: suggest change “uncovered” to “disclosed”.

Thank you for your suggestion. The word “uncovered” has been changed to “disclosed”. 

5. Line 25: change “enzymic” to “enzymatic”.

Thank you for your suggestion. The word “enzymic” has been changed to “enzymatic”. .

6. Line 28: suggest change “RLK-Pelle DLSV” to “RLK-Pelle_DLSV”

Thank you for your reminding. “RLK-Pelle DLSV” has been changed to “RLK-Pelle_DLSV”.

7. Line 36-37: Plant latin name should be italicized

Thank you for your reminding. The format of the latin names has been set as italic.

8. Line 48: change “no reports” to “no report”

Thank you for your reminding. It has been corrected now.

9. Line 49: change “increas” to “increase”, need to check the grammar and spelling throughout the manuscript

Thank you for your reminding. We feel sorry for our carelessness. We have corrected it and checked the spelling and grammar throughout the manuscript.

10. Line 51: delete “and increase the risk of phytotoxicity”

Thank you for your suggestion. It has been deleted.

11. Line 61: delete “herbicide”.

Thank you for your suggestion. It has been deleted.

12. Line 93-95: in table 1, need to check the dose of the treatment.

The doses are correct actually. In order to improve the curve-fitting effect, a higher dose and a lower dose were added except for the middle five doses, which were diluted at 1:2 ratio.

13. Line 112-122: Why did the authors not to study the mechanisms of spaying both mesosulfuron-methyl and mefenpyr-diethyl for comparison and concentrate only on seed dressing?

The present manuscript proposed to change the current spray method to seed dressing, so the focus was on the mechanism of seed dressing. A comparative study of spraying and seed dressing may be carried out in our further investigation.

14. Line 151 and after: change “ANVOA” to “ANOVA”.

We feel so sorry for our carelessness. It has been corrected. 

15. Line 167: in table 2, the ratio of GR50 should be listed in the table

Thank you for your suggestion. The ratios of GR50 have been listed in the table.

16. Line 205-237: For transcriptome analysis, common sequence results and gene annotations maybe omitted, the authors should focus on DEGs and their dynamics related to certain pathways like stress responses, transcription factors and protein kinases which were induced by mefenpyr-diethyl.

Thank you for your suggestion. We have omitted the unnecessary results.

17. Line 239-240: For consistency, gene name like UGTs and ABCs should change position with their definitions

Thank you for your suggestion. We have exchanged the position of the gene names with their definitions.

18. Line 278 and after: the discussion section should be concise and informative. For practical use of seed dressing in weed management, the authors need to analyze the common or difference between this study and other previous studies, based on the evaluation of effects, risks or potential usage in current advances. The authors could make an objective and rigorous conclusion.

Thank you for your constructive suggestions. We have referred to several other articles in the first paragraph of the discussion to compare the common or difference between this study and other previous studies. In addition, the agronomic value of seed dressing is also stressed and the conclusion part is reorganized.

19. Line 293: change “in” to “between”

Thank you for your reminding. It has been changed.

References:

Dias R de C, Bianchi L, Anunciato VM, Tropaldi L, Silva PV da, Carbonari CA, et al. Mefenpyr-diethyl as a safener for haloxyfop-methyl in bahiagrass. Ornam Hortic. 2021. doi:10.1590/2447-536x.v27i3.2306

Fritz BK, Hoffmann WC, Parker C, López JD. Development and testing of a laboratory spray table methodology to bioassay simulated levels of aerial spray drift. J ASTM Int. 2009;6: 1-10. doi:10.1520/JAI102125

Gao S, Han Y, Jiang L, Tao B. Study on bioassay method of fomesafen. J Northeast Agric Univ. 2011;42: 45–49. doi:10.3969/j.issn.1005-9369.2011.07.008

Senarathne SHS, Fernando RDV, Sangakkara UR. The allelopathic effects of Lantana camara on seed germination and growth of selected bioassay species. Korean J Weed Sci. 2011;31: 271–278. doi:10.5660/kjws.2011.31.3.271

Xie Y. Effects of safeners on the mesosulfuron tolerance of wheat cultivars (Triticum aestivum L.). China Agricult Univ. 2004. doi: 10.7666/d.y659620

Zhang C, Li X, Huang H, Wei S. Alert and prevention of the spreading of Aegilops tauschii, a worst weed in wheat field. J plant Prot. 2007;34: 103–106. doi:10.3321/j.issn:0577-7518.2007.01.020

We tried our best to improve the manuscript and made some changes in the manuscript. These changes will not influence the content and framework of the paper. And here we did not list the changes but marked in the revised paper.

We appreciate for Editors/Reviewers’ warm work earnestly, and hope that the correction will meet with approval.

Once again, thank you very much for your comments and suggestions. If you have any question, please contact us. I am looking forward to hearing from you.

Best regards and wishes!

Respectfully submitted,

Libing Yuan (Ph.D.)

College of Plant Protection, Hebei Agricultural University

Plant Protection Institute, Hebei Academy of Agricultural and Forestry Sciences, P.R. China

Tel. /Fax: +86-312-5915162.

E-mail:yuanlibing83@163.com

---

## [Decision Letter · Decision Letter 1]

2 Aug 2021

PONE-D-21-06771R1

Seed dressing with mefenpyr-diethyl as a safener for mesosulfuron-methyl application in wheat:the evaluation and mechanisms

PLOS ONE

Dear Dr. Yuan,

Thank you for submitting your manuscript to PLOS ONE. After careful consideration, we feel that it has merit but does not fully meet PLOS ONE’s publication criteria as it currently stands. Therefore, we invite you to submit a revised version of the manuscript that addresses the points raised during the review process.

We look forward to receiving your revised manuscript.

Kind regards,

Vijay Gahlaut, Ph.D.

Academic Editor

PLOS ONE

Journal Requirements:

Reviewers' comments:

Reviewer's Responses to Questions

**Comments to the Author**

1. If the authors have adequately addressed your comments raised in a previous round of review and you feel that this manuscript is now acceptable for publication, you may indicate that here to bypass the “Comments to the Author” section, enter your conflict of interest statement in the “Confidential to Editor” section, and submit your "Accept" recommendation.

Reviewer #2: All comments have been addressed

2. Is the manuscript technically sound, and do the data support the conclusions?

Reviewer #2: Yes

3. Has the statistical analysis been performed appropriately and rigorously? 

Reviewer #2: Yes

4. Have the authors made all data underlying the findings in their manuscript fully available?

Reviewer #2: Yes

5. Is the manuscript presented in an intelligible fashion and written in standard English?

Reviewer #2: Yes

6. Review Comments to the Author

Reviewer #2: The authors have made some corrections according to the editor and reviewers’ suggestions. I recommend it to be published and suggest minor revisions to improve the quality and readability.

Here are some revisions suggested:

Line 38: need to rephrase “yield loss of production in wheat producing regions”.

Line 147: pay attention to the format of “2-∆∆Ct”.

Line 172: delete “of plants”.

Line 226: lowercase for “L” in “alpha-Linolenic acid metabolism”.

Line 250: check the spelling “mephenpr-diethyl”.

Line 257: check the spelling “WORKY”.

Line 278: check the style of citation “Duhoux et al. (2017)” throughout the manuscript, which differ to line 59,100,118,300,318……

Line 283: suggest change “researches” to “studies”.

Line 320: need to rephrase the sentence “……could be evidence.”

Line 355: unified the citation of “S2 Fig” or “Fig S3”, and table?

Line 378: change “enzymic” to “enzymatic”.

7. PLOS authors have the option to publish the peer review history of their article (what does this mean?). If published, this will include your full peer review and any attached files.

Reviewer #2: No

---

## [Author Response · Author response to Decision Letter 1]

5 Aug 2021

Dear Editor and Reviewer:

Thank you for your letter and for the reviewer’s comments concerning our manuscript entitled “Seed dressing with mefenpyr-diethyl as a safener for mesosulfuron-methyl application in wheat: the evaluation and mechanisms.” (PONE-D-21-06771R1). Those comments are all valuable and very helpful for revising and improving our paper, as well as of important guiding significance to our researches. We have studied the comments carefully and made corrections which we hope meet with approval. Revised portions are marked under track changes mode in MS. The main corrections in the paper and the responds to the reviewer’s comments are as following:

Please NOTE: the red words are comments or questions from the editor or reviewer, and the black words are our replies.

Reviewer #2:

1.Line 38: need to rephrase “yield loss of production in wheat producing regions”.

Thanks for your suggestion. We have rephrased this sentence by deleting the unnecessary words.

2.Line 147: pay attention to the format of “2-∆∆Ct”

Thank you for your reminding. The “-∆∆Ct” has been set in upper superscrip format.

3.Line 172: delete “of plants”.

Thank you for your suggestion. The phrase “of plants” has been deleted to make the sentence concise.

4.Line 226: lowercase for “L” in “alpha-Linolenic acid metabolism”.

Thank you for your reminding. The word “Linolenic” has been changed to lowercase.

5.Line 250: check the spelling “mephenpr-diethyl”.

Thank you for your reminding. The word “mephenpr-diethyl” has been changed to “mefenpyr-diethyl” throughout the manuscript.

6.Line 257: check the spelling “WORKY”

Thank you for your reminding. “WORKY” has been changed to “WRKY”.

7.Line 278: check the style of citation “Duhoux et al. (2017)” throughout the manuscript, which differ to line 59,100,118,300,318……

Thank you for your reminding. The style of citation “Duhoux et al. (2017)” has been unified.

8.Line 283: suggest change “researches” to “studies”

Thank you for your suggestion. It has been changed now.

9.Line 320: need to rephrase the sentence “……could be evidence.”

Thank you for your reminding. We have rephrased this sentence.

10.Line 355: unified the citation of “S2 Fig” or “Fig S3”, and table?

Thank you for your reminding. We have checked all the format of the supplementary Figs and tables and unified them.

11.Line 378: change “enzymic” to “enzymatic”.

Thank you for your suggestion. It has been corrected.

We tried our best to improve the manuscript and made some changes in the manuscript, including those in the format of the references list. These changes will not influence the content and framework of the paper. And here we did not list the changes but marked in the revised paper.

We appreciate for Editors/Reviewers’ warm work earnestly, and hope that the correction will meet with approval.

Once again, thank you very much for your comments and suggestions. If you have any question, please contact us. I am looking forward to hearing from you.

Best regards and wishes!

Respectfully submitted,

Libing Yuan (Ph.D.)

College of Plant Protection, Hebei Agricultural University

Plant Protection Institute, Hebei Academy of Agricultural and Forestry Sciences, P.R. China

Tel. /Fax: +86-312-5915162.

E-mail:yuanlibing83@163.com

---

## [Editor Report · Decision Letter 2]

18 Aug 2021

Seed dressing with mefenpyr-diethyl as a safener for mesosulfuron-methyl application in wheat:the evaluation and mechanisms

PONE-D-21-06771R2

Dear Dr. Yuan,

We’re pleased to inform you that your manuscript has been judged scientifically suitable for publication and will be formally accepted for publication once it meets all outstanding technical requirements.

Kind regards,

Vijay Gahlaut, Ph.D.

Academic Editor

PLOS ONE
---

## [Editor Report · Acceptance letter]

20 Aug 2021

PONE-D-21-06771R2 

Seed dressing with mefenpyr-diethyl as a safener for mesosulfuron-methyl application in wheat: the evaluation and mechanisms. 

Dear Dr. Yuan:

I'm pleased to inform you that your manuscript has been deemed suitable for publication in PLOS ONE. Congratulations! Your manuscript is now with our production department. 

Kind regards, 

on behalf of

Dr. Vijay Gahlaut 

Academic Editor

PLOS ONE